# Pseudo-Label Training and Model Inertia in Neural Machine Translation

**Benjamin Hsu, Anna Currey, Xing Niu, Maria Nădejde and Georgiana Dinu**
AWS AI Labs
benhsu@amazon.com

## Abstract

Like many other machine learning applications, neural machine translation (NMT) benefits from over-parameterized deep neural models. However, these models have been observed to be brittle: NMT model predictions are sensitive to small input changes and can show significant variation across re-training or incremental model updates. This work studies a frequently used method in NMT, pseudo-label training (PLT), which is common to the related techniques of forward-translation (or self-training) and sequence-level knowledge distillation. While the effect of PLT on quality is well-documented, we highlight a lesser-known effect: PLT can enhance a model's stability to model updates and input perturbations, a set of properties we call *model inertia*. We study inertia effects under different training settings and we identify distribution simplification as a mechanism behind the observed results.

## 1 Introduction

Self-training (Fralick, 1967; Amini et al., 2022) is a popular semi-supervised technique used to boost the performance of neural machine translation (NMT) models. In self-training for NMT, also known as forward-translation, an initial model is used to translate monolingual data; this data is then concatenated with the original training data in a subsequent training step (Zhang & Zong, 2016; Marie et al., 2020; Edunov et al., 2020; Wang et al., 2021). Self-training is believed to be effective through inducing input smoothness and leading to better learning of decision boundaries from the addition of unlabeled data (Chapelle et al., 2006; He et al., 2020; Wei et al., 2021). It has also been observed to effectively diversify the training distribution (Wang et al., 2021; Nguyen et al., 2020).

A closely related technique is that of knowledge distillation (Hinton et al., 2015; Gou et al., 2021), particularly *sequence-level* knowledge distillation (SKD), which uses hard targets in training and reduces to pseudo-labeled data augmentation (Kim & Rush, 2016). In NMT, knowledge distillation is effective through knowledge transfer from ensembles or larger-capacity models and as a data augmentation method (Freitag et al., 2017; Gordon & Duh, 2019; Tan et al., 2019; Currey et al., 2020). In non-autoregressive translation, Zhou et al. (2020) explored the effect of SKD on training data complexity and showed that simpler training data from distillation is crucial for the performance of non-autoregressive MT models.

This paper examines the component that is common to these techniques, the introduction of pseudo-labeled training (PLT) data. We focus on the more common autoregressive NMT formulation and show that in addition to the known quality gains, PLT has a large impact on model brittleness in that it increases smoothness as well as stability across model re-training. Our main contributions are:

- We focus on a set of stability properties in NMT models, which we unify under the umbrella term *inertia*, and show that PLT increases model inertia. We further show that both the quality gains and the improved inertia are not properties of any one specific technique such as self-training or knowledge distillation, but are common to the use of pseudo-labeled data in training.

- We investigate the hypothesis that the observed properties correlate with a training data simplification mechanism, similarly to the observations made in Zhou et al. (2020). We compare with other popular semi-supervised techniques to investigate if the model quality and inertia properties hold when distribution simplification effects are not present.

- Based on our findings, we recommend incorporating PLT into NMT training whenever inertia (e.g., stability to input perturbations and across incremental model updates) is important, as it increases inertia without sacrificing quality.

## 2 RELATED WORK

Neural network models are known to be sensitive to input variations, i.e., lacking in *smoothness*. This can make them brittle or open to adversarial attacks, a property observed across many application domains (Goodfellow et al., 2014; Szegedy et al., 2014; Jia & Liang, 2017). Neural machine translation models are similarly prone to robustness issues and can be affected by both synthetic and natural noise, leading to lower translation quality (Belinkov & Bisk, 2018; Li et al., 2019; Niu et al., 2020; Fadaee & Monz, 2020). In MT, earlier works have found noisy data augmentation (Belinkov & Bisk, 2018) and subword regularization (Kudo, 2018; Provilkov et al., 2020) to be among the most simple yet effective methods for addressing instability to input perturbations.

In addition to smoothness, neural models are known be sensitive to the various sources of randomness in training, such as initialization or dropout (Bengio, 2012; Reimers & Gurevych, 2017; Madhyastha & Jain, 2019). This instability negatively impacts end-users in the form of spurious differences in outputs between model updates, or more acutely, as quality regressions on specific data points, also known as negative flips (Shen et al., 2020; Xie et al., 2021; Yan et al., 2021). In NLP, Cai et al. (2022) focus on a set of structured prediction tasks and show that when random initialization changes, up to 30% of all errors can be regression errors, and that improved accuracy does not always mean reduced regressions. While negative flips are more difficult to measure in MT as multiple translations can be valid, the lack of consistency across re-training is a known problem: in our experiments ∼80% of the translations change due to different model random initialization alone. Despite this, to the best of our knowledge, minimizing regressions or improving stability across incremental model updates or re-trainings has not yet been addressed in MT.

This paper examines pseudo-label training in NMT and its effect on stability to both input variations and incremental model updates, which we group under the term *inertia*. Earlier work on pseudo-label training in MT focused on measuring quality alone and did not shed light on stability-related properties (Wang et al., 2021; He et al., 2020; Wei et al., 2021; Yuan et al., 2020). In terms of stability to input variations, or smoothness, our findings are related to the work of Papernot et al. (2015), where authors introduce *defensive distillation* and show that (self-)distillation increased smoothness when tested on digit and object recognition tasks. They show that the effect is one of reducing the amplitude of the network gradients. Unlike our work, they do not test pseudo-label training, but soft-target distillation, where a student is trained using the prediction probabilities of a teacher.

Finally, we hypothesize that PLT techniques are able to increase model inertia based on their distribution simplification properties. Earlier works have explored the distribution simplification property of PLT methods in terms of model performance. In non-autoregressive NMT, Zhou et al. (2020) and Xu et al. (2021) explored the effect of SKD on training data *complexity* and its correlation with model performance. As in previous work, they hypothesized that SKD alleviates the multiple modes problem, i.e., the existence of multiple alternative translations (Gu et al., 2018). Similarly to Zhou et al. (2020), we measure training data complexity when adding pseudo-labeled data and use the entropy of a conditional word-level alignment as a complexity metric.

## 3 TRAINING WITH PSEUDO-LABELS IN NMT

**Neural machine translation (NMT)**    We use the autoregressive formulation of NMT, where given parallel data containing source and target sequences, a model $\theta$ is learned using the following objective:

$$\mathcal{L} = -\sum_{j=1}^{J} \sum_{k=1}^{|V|} 1\{y_j = k\} \times \log p(y_j = k | \mathbf{y}_{<\mathbf{j}}, \mathbf{x}, \theta), \tag{1}$$

where $\mathbf{x} = [\mathbf{x_1}, ..., \mathbf{x_I}]$ and $\mathbf{y} = [\mathbf{y_1}, ..., \mathbf{y_J}]$ are the source/target sequences respectively, $I$ and $J$ are the source/target length, and $|V|$ is the size of the vocabulary. Unless otherwise stated, we use beam search with a fixed number of hypotheses in order to generate a translation from this model.

**Pseudo-label training (PLT)**   In this paper, we introduce the term *pseudo-label training* (PLT) to refer to the general technique of adding pseudo-labeled data during training, where the labels are obtained using a previously trained NMT model. Specifically, we consider two-step PLT. In a first stage we estimate a *teacher* model $\theta^*$ trained with a supervised loss on samples drawn from $p$, the empirical distribution of the original training data:

$$\mathcal{L} = -\mathbb{E}_{x \sim p(x)} \mathbb{E}_{y \sim p(y|x)} p(y|x) \, log \, p_{\theta^*}(y|x)$$

In a second step we estimate the final *student* model $\theta$, combining the supervised loss with a PL (pseudo-label) loss $\mathcal{L} + \mathcal{L}_{PL}$, where:

$$\mathcal{L}_{\mathcal{PL}} = -\mathbb{E}_{x \sim p^{PL}(x), y'} \, log \, p_\theta(y'|x)$$

In this case the targets $y'$ are given by the teacher distribution $p_{\theta^*}$ and the samples are drawn from a second distribution, $p^{PL}$, which varies in the experiments below.

**Related techniques**   As discussed earlier, PLT is a common feature of several widely used techniques in NMT such as self-training (a.k.a. forward-translation) and sequence-level knowledge distillation. This paper opts for the term *pseudo-label training (PLT)* in order to avoid confusion with additional assumptions made by these techniques. Specifically:

- PLT does not necessarily imply semi-supervision, as self-training does.
- PLT is more specific than KD in that it is restricted to hard labels (as opposed to training on soft targets as in Hinton et al. 2015), but more generic as it does not assume model compression.

Another technique for introducing synthetic data is the use of back-translation (BT), where target segments are translated into source segments (Sennrich et al., 2016a; Hoang et al., 2018; Edunov et al., 2020). PLT does not include BT since the latter does not introduce synthetic *targets* or labels.

Lastly, note that self-training is closely related to entropy minimization (Grandvalet & Bengio, 2004), a semi-supervised technique that encourages high-confidence predictions on unlabeled data. When reducing this objective to its mode, it becomes identical to $\mathcal{L}_{\mathcal{PL}}$ above, also observed in He et al. (2020).

## 4   MODEL INERTIA

This section introduces a set of desired stability-related MT properties that we group under the term *inertia*. All our metrics are closed-box (based on user-*observed* model behaviour alone) and we investigate two types of model inertia: (1) robustness to input perturbations (or smoothness) and (2) stability across incremental model updates.

### 4.1   INPUT SMOOTHNESS

Robustness to input variations is important in MT models, which have been shown to be negatively affected by misspellings and other small variations in input (Belinkov & Bisk, 2018). Niu et al. (2020) introduced metrics that contrast translations of noisy input with those of their clean counterparts in order to disentangle robustness from generic quality changes. We evaluate model robustness and consistency to input changes following the definitions introduced in Niu et al. (2020): `Robustness` measures degradation in translation *quality* when small variations are present in the input, while `Consistency` is a reference-free metric for changes in translation *output* alone. Specifically:

$$\begin{aligned} \texttt{Consistency} &= \text{H}(\text{BLEU}(Y', Y), \text{BLEU}(Y, Y')) \\ \texttt{Robustness} &= \text{BLEURT}(Y', Y_{ref}) - \text{BLEURT}(Y, Y_{ref}) \end{aligned} \quad (2)$$

where $Y_{ref}$ stands for reference translations, $Y, Y'$ are translations of a *clean/noisy* versions of the test set (e.g., one with introduced misspellings) and $\text{H}(\cdot, \cdot)$ stands for the harmonic mean. In this paper, we expand these definitions to consider robustness not only to synthetic misspellings, but also to natural grammatical errors.

## 4.2 STABILITY TO MODEL UPDATES

Unlike smoothness metrics, stability metrics are functions of *two* models: an original one (e.g., one that is deployed and available to users) and an update of this model which implements an incremental change. We denote a model update as a pair $(\theta, D, A)_i$, $(\theta, D, A)_{i+1}$, where $\theta$ are the model parameters obtained when training using data $D$ and algorithm $A$. While many incremental updates are possible, in this work we keep the model size and architectures intact and vary the random parameter initialization in training, following Xie et al. (2021) and Cai et al. (2022). We define *stability* as a measure of similarity between model outputs, irrespective of quality changes, while *regressions* (negative flips) measure output changes that result in lower quality on a given input segment (Cai et al., 2022; Xie et al., 2021; Shen et al., 2020; Yan et al., 2021).

**STABILITY** Stability is measured as string similarity between the different outputs $Y_i, Y_{i+1}$. We use a symmetric BLEU-based metric, the harmonic mean between $\text{BLEU}(Y_i, Y_{i+1})$ and $\text{BLEU}(Y_{i+1}, Y_i)$, where $Y_i$ and $Y_{i+1}$ are translations obtained with models $\theta_i$ and $\theta_{i+1}$, respectively.

**NFR** Similarly to earlier works (Cai et al., 2022; Xie et al., 2021; Yan et al., 2021), we measure regressions as **N**egative **F**lip **R**ate, the number of sentences for which the translation degrades between model updates over the total number of segments. We consider degradations in terms of both overall quality and a targeted translation error category. Unlike other tasks, NMT lacks a reliable automatic *segment-level* quality metric (Kocmi et al., 2021; Mathur et al., 2020); we use human evaluations for this reason. Having an additional targeted error category allows us to measure segment-level regression automatically. In this work, we adopt gender translation accuracy as the targeted error category.

**NFI** Following Cai et al. (2022), we also measure regressions in terms of **N**egative **F**lip **I**mpact. NFI is defined as the proportion of negative flips to the total number of errors made by the new model. Note that in NMT, error is less well-defined for quality since it is not a categorical concept. This is not the case with targeted translation error categories.

## 5 EXPERIMENTS

We perform experiments across 6 language pairs (LPs): English (en)↔German (de), Russian (ru), and Japanese (ja). We adapt the Transformer-base architecture (Vaswani et al., 2017) to 20 encoder layers and 2 decoder layers (denoted *20:2*) as recommended by Domhan et al. (2020) and SSRU decoder layers for faster decoding (Kim et al., 2019). The deep-encoder-shallow-decoder configuration is widely used (Miceli Barone et al., 2017; Kim et al., 2019; Kasai et al., 2021), and the 20:2 model was found by Domhan et al. (2020) to yield comparable quality to the 6:6 and 10:10 models while significantly decreasing latency. Unless otherwise noted, we use beam decoding with a beam size of 5 (further details in Appendix A).

Experiments are carried out with the WMT21 dataset (Akhbardeh et al., 2021). For en↔de we use 286M parallel segments, for en↔ja we use 17.2M parallel segments, and for en↔ru we use 34M parallel segments. For development, we use WMT newstest datasets from earlier years (see Appendix B for more details on datasets used). We evaluate quality using BLEU[1] (Papineni et al., 2002) and BLEURT (Sellam et al., 2020) on the WMT21 newstest sets (Akhbardeh et al., 2021). We use only source-original test sets in order to avoid misestimating model performance due to translationese input (Marie et al., 2020).

We train PLT-augmented models using a mix of the original training data and pseudo-labeled data in a joint training setting following Zhang & Zong (2016); Gordon & Duh (2019). Based on recommendations by He et al. (2020), we use dropout for all the models, set to 0.1. We do not tune the trade-off between the two losses $\mathcal{L}$ and $\mathcal{L}_{\mathcal{PL}}$ (we use an equal amount of original and PLT data) or the number of incremental applications of the PLT augmentation.

---

[1]Specifically, using `sacreBLEU` (Post, 2018) with signature: `nrefs:1|case:mixed|eff:no|tok:13a|smooth:exp|version:2.0.0` except for en→ja where we use the `ja-mecab` tokenizer.

| SRC | Thousands of people **aree guven** a drug and thousands of others are given a placebo.**.** |
| BASELINE | Tausende von Menschen erhalten **Guven** ein Medikament und Tausende von anderen erhalten ein Placebo. |
| PLT(TRAIN) | Tausende von Menschen erhalten eine Droge und Tausende von anderen erhalten ein Placebo. |
| SRC | Can yo put cites on those? |
| BASELINE | Können Sie Zitate darauf setzen? |
| PLT(TRAIN) | Kannst du diese zitieren? |

Table 1: Example translations from BASELINE and PLT(TRAIN) on the synthetic misspellings and GMEG test sets. In the first example (synthetic misspelling), the baseline invents the word *Guven* as a translation of the original miss-spelled word, *guven(given)*. PLT translates the second example (English learner error) as *Can you cite these?* using the informal register, while the Baseline translates it literally as *Can you put citations on these?* (formal register).

## 5.1 QUALITY AND INERTIA USING PSEUDO-LABELED DATA

This section evaluates PLT for both generic model quality and for inertia. Unless otherwise noted, student models share the same architecture as the teacher and are trained using the same parallel data with the addition of pseudo-labeled data. PLT can be implemented by sampling and labeling data from different source distributions $p^{PL}$: the original training data (as in KD) or unseen monolingual data (i.e. semi-supervised). This section tests both: to that end, teacher models are trained on *half* of the available parallel data, while the other half is reserved as a source of unlabeled monolingual data. Specifically, we compare:

- BASELINE: Model trained on half the available data without any data augmentation.
- PLT(TRAIN): Data used in PLT augmentation is sampled from the training data.
- PLT(UL): Data used in PLT augmentation is sampled from unused parallel data.
- ALLDATA: Finally, to account for the differences in training data size, we also compare against a model trained on *all* available parallel data without any PLT.

### 5.1.1 INPUT SMOOTHNESS

For each of these models we compute newstest quality (BLEU score) as well as model smoothness (robustness and consistency). We measure robustness and consistency as defined in Section 4 with the following sources of input variations:

- Synthetic misspellings: We introduce misspellings as proposed by Niu et al. (2020) into the newstest set. Each word is misspelled with probability of 0.1, and the strategy is randomly chosen from single-character deletion, insertion, and substitution (Karpukhin et al., 2019).
- GMEG: The GMEG corpus (Napoles et al., 2019) contains data with natural errors made by English language learners (grammatical misuse, misspellings, etc.). We compute consistency using the noisy input and a reference correction made by a professional annotator. We report the average consistency over the four provided reference corrections.[2]

Example translations and results are show in Tables 1 and 2, respectively. Across all LPs, translation quality improves when pseudo-labeled data is used in training, irrespective of the source of the data added. However, sampling from unseen data does not bring additional improvements over using seen data for PLT. Similarly, using all parallel data vs. only half is not beneficial across the board, suggesting limitations of the training data w.r.t. the test domain.

PLT shows significantly higher model consistency on both synthetic misspellings and the GMEG test sets.[3] Unlike Niu et al. (2020), however, we find that robustness scores (translation *quality* changes relative to input changes) are not as well correlated with consistency scores, suggesting that while translations are more stable under noisy conditions they may not necessarily be better. In the context

---

[2] It is not possible to compute robustness scores for GMEG as this set does not contain reference translations.

[3] The noisy datasets (synthetic misspellings and GMEG) do not cover the ja→en translation direction, so this direction is not included in Table 2.

| LP | Setting | Teacher | Student (Original + PL) | BLEU | BLEURT | Misspellings | | GMEG |
| | | | | | | Rob | Const | Const |
|---|---|---|---|---|---|---|---|---|
| en→de | ALLDATA | – | 286M + 0 | $27.83_{\pm1.1}$ | $-0.132_{\pm0.03}$ | $-0.725_{\pm0.04}$ | $73.9_{\pm0.9}$ | 83.1 |
| | BASELINE | – | 143M + 0 | $27.62_{\pm1.1}$ | $-0.126_{\pm0.03}$ | $\mathbf{-0.684}_{\pm0.04}$ | $73.1_{\pm0.9}$ | 82.8 |
| | PLT(TRAIN) | 143M | 143M + 143M (Train) | $27.86_{\pm1.1}$ | $\mathbf{-0.115}_{\pm0.03}$ | $-0.802_{\pm0.04}$ | $76.5_{\pm0.8}$ | 85.1 |
| | PLT(UL) | 143M | 143M + 143M (UL) | $\mathbf{28.07}_{\pm1.1}$ | $-0.118_{\pm0.03}$ | $-0.779_{\pm0.04}$ | $\mathbf{76.8}_{\pm0.8}$ | $\mathbf{85.3}$ |
| en→ru | ALLDATA | – | 34M + 0 | $25.71_{\pm1.1}$ | $0.027_{\pm0.03}$ | $\mathbf{-0.774}_{\pm0.05}$ | $66.4_{\pm1.1}$ | 79.3 |
| | BASELINE | – | 17M + 0 | $25.08_{\pm1.1}$ | $0.035_{\pm0.03}$ | $-0.893_{\pm0.05}$ | $64.9_{\pm1.0}$ | 78.4 |
| | PLT(TRAIN) | 17M | 17M + 17M (Train) | $\mathbf{26.16}_{\pm1.1}$ | $\mathbf{0.044}_{\pm0.03}$ | $-0.901_{\pm0.05}$ | $\mathbf{70.2}_{\pm1.0}$ | 81.5 |
| | PLT(UL) | 17M | 17M + 17M (UL) | $25.87_{\pm1.1}$ | $0.044_{\pm0.03}$ | $-0.920_{\pm0.05}$ | $70.1_{\pm1.0}$ | $\mathbf{82.2}$ |
| en→ja | ALLDATA | – | 17.2M + 0 | $23.62_{\pm0.8}$ | $-0.289_{\pm0.03}$ | $\mathbf{-0.834}_{\pm0.05}$ | $59.3_{\pm1.1}$ | 72.6 |
| | BASELINE | – | 8.6M + 0 | $22.82_{\pm0.9}$ | $-0.316_{\pm0.03}$ | $-0.844_{\pm0.05}$ | $59.3_{\pm1.1}$ | 71.5 |
| | PLT(TRAIN) | 8.6M | 8.6M + 8.6M (Train) | $24.63_{\pm0.9}$ | $-0.280_{\pm0.03}$ | $-0.959_{\pm0.06}$ | $\mathbf{64.5}_{\pm1.1}$ | $\mathbf{75.9}$ |
| | PLT(UL) | 8.6M | 8.6M + 8.6M (UL) | $\mathbf{24.59}_{\pm0.9}$ | $\mathbf{-0.271}_{\pm0.03}$ | $-0.905_{\pm0.05}$ | $\mathbf{64.5}_{\pm1.1}$ | $\mathbf{75.9}$ |
| de→en | ALLDATA | – | 286M + 0 | $32.32_{\pm1.1}$ | $0.393_{\pm0.03}$ | $\mathbf{-1.202}_{\pm0.06}$ | $77.4_{\pm1.0}$ | - |
| | BASELINE | – | 143M + 0 | $32.36_{\pm1.1}$ | $0.394_{\pm0.03}$ | $-1.246_{\pm0.06}$ | $76.9_{\pm1.0}$ | - |
| | PLT(TRAIN) | 143M | 143M + 143M (Train) | $\mathbf{32.93}_{\pm1.1}$ | $0.397_{\pm0.03}$ | $-1.300_{\pm0.06}$ | $80.0_{\pm0.8}$ | - |
| | PLT(UL) | 143M | 143M + 143M (UL) | $32.50_{\pm1.1}$ | $\mathbf{0.400}_{\pm0.03}$ | $-1.320_{\pm0.06}$ | $\mathbf{80.2}_{\pm0.9}$ | - |
| ru→en | ALLDATA | – | 34M + 0 | $36.39_{\pm1.2}$ | $0.324_{\pm0.03}$ | $\mathbf{-0.363}_{\pm0.04}$ | $85.6_{\pm0.8}$ | - |
| | BASELINE | – | 17M + 0 | $36.26_{\pm1.2}$ | $0.321_{\pm0.03}$ | $-0.442_{\pm0.04}$ | $84.2_{\pm0.9}$ | - |
| | PLT(TRAIN) | 17M | 17M + 17M (Train) | $\mathbf{37.00}_{\pm1.2}$ | $\mathbf{0.337}_{\pm0.03}$ | $-0.391_{\pm0.04}$ | $\mathbf{87.6}_{\pm0.7}$ | - |
| | PLT(UL) | 17M | 17M + 17M (UL) | $36.85_{\pm1.12}$ | $0.329_{\pm0.03}$ | $-0.430_{\pm0.04}$ | $87.0_{\pm0.7}$ | - |

Table 2: Training data sizes and performance scores for PLT/Baseline models. Quality is measured with BLEU and BLEURT on the WMT21 newstest set. Smoothness is measured as robustness and consistency to synthetic (**Misspellings**) and natural (**GMEG**) noise. GMEG scores are computed as the average over four reference corrections. Robustness measures changes in translation quality w.r.t. input variations, while consistency measures translation *changes* alone.

| Setting | en→de | | de→en | | en→ja | | ja→en | | en→ru | | ru→en | |
| | St. | EM | St. | EM | St. | EM | St. | EM | St. | EM | St. | EM |
|---|---|---|---|---|---|---|---|---|---|---|---|---|
| ALLDATA | 73.32 | 15.6% | 77.45 | 28.8% | 54.56 | 2.1% | 44.73 | 4.6% | 63.45 | 7.9% | 69.78 | 14.1% |
| BASELINE | 72.48 | 14.2% | 77.95 | 30.4% | 53.64 | 2.1% | 40.78 | 3.6% | 60.48 | 7.4% | 67.01 | 11.3% |
| PLT-$\delta$(STUDENT) | **82.03** | **27.9%** | **86.34** | **46.3%** | **65.12** | **5.6%** | **56.89** | **8.4%** | **72.93** | **14.3%** | **77.11** | **24.8%** |
| PLT-$\delta$(TEACHER) | 81.45 | 26.3% | 85.04 | 42.4% | 63.25 | 5.5% | 54.75 | 5.6% | 69.97 | 11.3% | 74.81 | 20.6% |
| DISTIL. | 75.44 | 16.2% | 80.57 | 35.3% | 44.73 | 4.6% | 44.54 | 3.5% | 64.51 | 7.0% | 70.20 | 15.0% |

Table 3: Stability (*St.*) to model updates, in this case re-training with different random seed. Exact match (*EM*) is the percent of outputs that stay identical across the two models. For Distillation (*Distil.*), the second model is trained to mimic the first model.

of semi-supervised learning, it has been hypothesized that self-training has the effect of making models smoother through the addition of new data (He et al., 2020; Wei et al., 2021). Our results suggest that this is not necessarily the case, as smoothness results are similar irrespective of the use of new unlabeled (monolingual) data (i.e., PLT(TRAIN) and PLT(UL) have similar smoothness).

### 5.1.2 STABILITY TO MODEL UPDATES

Next we investigate stability properties with respect to model *updates* when PLT is used in training. We fix the source of the pseudo-labeled data to be the training data (i.e., we consider only PLT(TRAIN)) and compare translation changes when re-training a model. Recall, a model update consists of a pair $(\theta, D, A)_1, (\theta, D, A)_2$, where $\theta$ are the model parameters obtained when training using data $D$ and algorithm $A$. In these experiments, we keep the network architecture identical and hold $A_1 = A_2$, modulo the random seed used in initialization. We contrast several settings:

- BASELINE: Models are trained and re-trained with *half* the original data ($D_1 = D_2$), and *no* pseudo-labeled data is used. As above, we also evaluate the case where *all* of the original data is used (ALLDATA). We vary the random seed, leading to $\theta_1 \neq \theta_2$.

- PLT-$\delta$(STUDENT): This tests the hypothesis that using PLT leads to more stable models that behave similarly when varying minor training conditions. We consider an identical setup as the baseline ($D_1 = D_2$), except that the data is augmented to contain PLT data

- PLT-$\delta$(TEACHER): In this setting, the two models $\theta_1$ and $\theta_2$ use PLT data in training; however, two different *teachers* are used to create it (the teachers are trained with different random seeds). This simulates a realistic setting where the teachers used to create pseudo-labels are not likely to

| | WMT21 | | | WinoMT | | | |
| | en→de | en→ja | en→ru | en→de | | en→ru | |
| Setting | NFR | NFR | NFR | NFR | NFI | NFR | NFI |
|---|---|---|---|---|---|---|---|
| ALLDATA | 18.1% | 16.5% | 14.1% | 4.7% | 14.3% | 5.4% | 9.2% |
| BASELINE | 18.0% | 14.9% | 16.1% | 5.2% | 17.0% | 5.4% | 9.0% |
| PLT-δ(STUDENT) | **7.4%** | 15.0% | **10.2%** | **3.2%** | **9.8%** | **3.2%** | **5.2%** |
| PLT-δ(TEACHER) | 7.8% | **13.6%** | 11.1% | 3.6% | 10.8% | 3.2% | 5.2% |
| DISTIL. | 14.4% | 19.1% | 10.9% | 4.7% | 14.8% | 4.4% | 7.2% |

Table 4: Negative flip rate (NFR) and negative flip impact (NFI) on WMT21 (assessed by human annotators) and WinoMT (using the automatic gender translation accuracy metric).

stay constant. Note however that this is not a direct comparison to the baseline and PLT methods: the models do not vary in random seed alone, but also the contents of the training data ($D_1 \neq D_2$).

- DISTILLATION: In this setting $D_2$ is obtained from $D_1$ using pseudo-labeled data obtained with model $\theta_1$. The training data $D_1$ is re-translated and merged with the original $D_1$ to create $D_2$. This setting is a standard distillation approach for minimizing regressions, where $\theta_2$ is trained to explicitly mimic $\theta_1$'s predictions (Yan et al., 2021; Cai et al., 2022).

Stability and regression metrics are averaged over $(\theta_1, \theta_2)$ and $(\theta_2, \theta_1)$ scores since random initialization changes are not directional model updates. Tables 3 and 4 show stability and regression metrics respectively (regression results are discussed in the next section).

First, we observe that a striking number of translations change when changing random initialization: only 15% of outputs remain identical for en→de, and 8% and 2% remain identical for the lower-resource en→ru and en→ja pairs respectively. Doubling the amount of training data (ALLDATA) improves stability, but not by a large margin. Across all LPs tested, PLT improves stability relative to the baseline models and nearly doubles the percentage of segments translated identically. Interestingly, PLT also improves stability relative to the system trained on all available parallel data, once again indicating that inertia effects do not simply stem from more data. This result is particularly surprising for the PLT-δ(TEACHER) setting: unlike the baseline, the two models compared are trained on *different* data on the target side, yet their outputs are more similar to each other than the baseline outputs are to each other. This suggests that the high translation variability of the original data (a.k.a. multiple modes in Gu et al., 2018) is an issue with auto-regressive MT as well, and that pseudo-labeled data alleviates it even when created with different models.

Finally, we also find that distillation, where a new model is explicitly trained to mimic the previous model, increases stability between teacher and student, confirming earlier observations on text classification Cai et al. (2022). However, this improvement is modest in our experiments.

### 5.1.3 NEGATIVE FLIPS

Next, we assess PLT in terms of negative flips (NFs) as described in Section 4. We evaluate regressions in terms of overall quality (human evaluations on the WMT21 newstest set) and on a targeted error category (gender translation accuracy). For human evaluations, we used two professional annotators who assigned scores on a scale of 1 to 6 with 0.2 increments, where 6 indicates a perfect translation. A NF is defined as both annotators agreeing that there is a degradation. Since quality is evaluated on a scale, and not as a binary score, the concept of NFI is ambiguous. We therefore compute negative flip rate (NFR) alone. We evaluate on en→de,ja,ru due to availability of annotators.

For gender translation accuracy, which aggregates categorical measurements, we evaluate both NFR and NFI. We use the WinoMT benchmark (Stanovsky et al., 2019), a gender accuracy benchmark with a reliable segment-level metric suitable for automatic measurements of negative flips. The dataset consists of English source segments containing a profession whose gender is ambiguous at the lexical level but disambiguated in the sentential context, along with an automatic morphology-based classifier that evaluates the gender accuracy when translated. We evaluate on the two of our language pairs that are covered by WinoMT, en→de and en→ru.

Results are shown in Table 4. First, we observe that, like other NLP tasks, regressions are also an issue for NMT: on WMT21, 15%-20% of the test set samples are judged as having degraded in quality according to both human annotators. NFR is lower for the gender translation accuracy task; however, NFs still amount to 10%-15% of the total number of errors, as measured by NFI. Mirroring

| LP | Teacher | | | PLT | | | | | |
|---|---|---|---|---|---|---|---|---|---|
| | Arch. | BLEU | BLEURT | BLEU | BLEURT | Stability | Const(GMEG) | Rob(Missp) | Const(Missp) |
| en→X | – | – | – | 25.12 | -0.136 | 62.20 | 77.57 | **-0.807** | 65.77 |
| | 20:1 | 24.08 | -0.185 | 25.54 | -0.083 | 72.67 | 79.93 | -0.876 | 69.72 |
| | 20:2 | 25.12 | -0.136 | 26.12 | **-0.043** | **74.10** | 80.72 | -0.868 | **70.38** |
| | 20:4 | **25.63** | **-0.136** | **26.27** | -0.044 | 73.68 | **80.87** | -0.886 | 70.05 |
| X→en | – | – | – | 28.12 | 0.076 | 61.91 | – | -0.844 | 80.53 |
| | 20:1 | 26.74 | -0.047 | 28.30 | 0.029 | 73.15 | – | -0.825 | 83.56 |
| | 20:2 | 28.12 | 0.076 | 29.56 | 0.108 | **73.96** | – | -0.875 | 83.80 |
| | 20:4 | **28.40** | **0.095** | **29.94** | **0.113** | 73.86 | – | **-0.808** | **83.99** |

Table 5: PLT models using teachers of varying quality (averages over the three language pairs in each direction). We find that teacher quality correlates with quality of PLT; however, weaker teachers can still improve student quality. In terms of inertia properties, these are preserved regardless of teacher quality. Note X→en averages for robustness and consistency to misspelling include only de,ru→en.

stability results, both PLT models have significantly fewer segment-level regressions than baseline models. For quality, this is most pronounced for en→de,ru (∼50%-100% relative NFR reduction). In contrast, the effect of distillation is not consistent across the language pairs or the two test sets.

## 5.2 TEACHER QUALITY

In previous sections, we found that quality and model inertia improved when using PLT regardless of the source of the data. In this section, we examine another dimension which distinguishes different flavors of PLT, namely teacher quality. Stronger teachers (teachers with larger capacity than the student) are more common in KD applications whereas identical teacher and student models are the norm in self-training/forward translation. Specifically, we vary the base 20:2 teacher architecture by decreasing the number of decoder layers to 1 (weaker teacher) and increasing it to 4 (stronger teacher). We keep the student architecture identical at 20:2 layers and fix the source of pseudo-labeled data to the training set (referred to as PLT(TRAIN) in earlier sections).

Interestingly, we find that teacher quality does not play a large role in model stability (Table 5). There are small improvements in stability and robustness when stronger teachers are used, but gains are in range for all teacher models considered, even for weak teachers. Stronger teachers, however, are responsible for better performing student models. Most surprisingly, we found quality improvements over the baseline even when the teacher is of worse quality than the baseline model. This corroborates other work suggesting that the mechanism behind PLT is not simply that of compressing better performing (sets of) models (Furlanello et al., 2018; Hahn & Choi, 2019; Yuan et al., 2020).

## 6 DISTRIBUTION SIMPLIFICATION

The previous section showed that PLT increases both quality and model inertia under different monolingual data and teacher quality settings. We hypothesize that the increased inertia observed is correlated with a distribution simplification mechanism: PLT leads to simpler training data, resulting in models that are less brittle. We test this by comparing PLT with other techniques used to improve quality and smoothness, but that may not have a distribution simplification effect. Below, we fix the source of pseudo-labeled data to the training data and test:

- BT: Back-translation, a commonly used semi-supervised method that adds parallel data obtained through translation of target data with a reverse-direction MT model.

- BPE-DROPOUT: A regularization method that has been shown to improve robustness to noise (Provilkov et al., 2020). We used a dropout rate of 0.1 as recommended by the authors.

- PLT(SAMPLE): A variant of PLT where we vary the decoding strategy and perform sampling decoding which leads to more complex data and weaker student models (Zhou et al., 2020). Specifically, we sampled the top-8 hypotheses.

In previous work, Zhou et al. (2020) proposed a conditional entropy measure of training data complexity and showed that non-autoregressive translation performance is dependent on simpler training data distributions, such as those obtained with SKD. Here, we use the same entropy-based measure.

| LP | Setting | $C(d)\downarrow$ | BLEU | BLEURT | Stability | Const(GMEG) | Rob(Missp) | Const(Missp) |
|---|---|---|---|---|---|---|---|---|
| en→X | BASELINE | 3.74 | 25.12 | -0.136 | 62.20 | 77.57 | -0.807 | 65.77 |
| | BT | 3.64 | 25.96 | -0.140 | 65.36 | 77.16 | -0.934 | 64.83 |
| | BPE-DROPOUT | 3.90 | 24.82 | -0.154 | 61.70 | 78.69 | **-0.547** | **71.86** |
| | PLT(SAMPLE) | 3.56 | 25.96 | -0.119 | 71.97 | 80.41 | -0.865 | 69.87 |
| | PLT(TRAIN) | **3.54** | **26.12** | **-0.117** | **74.10** | **80.73** | -0.868 | 70.38 |
| X→en | BASELINE | 3.12 | 28.12 | 0.076 | 61.91 | – | -0.844 | 80.53 |
| | BT | 3.01 | 28.88 | 0.099 | 64.91 | – | -0.979 | 80.00 |
| | BPE-DROPOUT | 3.41 | 28.19 | 0.075 | 61.73 | – | **-0.591** | **84.96** |
| | PLT(SAMPLE) | 2.98 | 29.41 | 0.107 | 72.69 | – | -0.849 | 83.70 |
| | PLT(TRAIN) | **2.94** | **29.56** | **0.108** | **73.96** | – | -0.875 | 83.80 |

Table 6: Quality and model inertia with PLT versus other methods (averages over the three language pairs in each direction). Stability to model updates is computed w.r.t. to random seed variation in student models. X→en averages for robustness and consistency to misspellings involve de,ru→en.

For each setting, we (1) compute an alignment model on the training data using `fast_align` (Dyer et al., 2013), (2) use it to align a sample of the training corpus, and (3) compute the entropy of the aligned data, leading to: $C(d) = -\frac{1}{|V_x|}\sum_{x\in V_x}\mathbb{E}_{y|x_{align}}log(y|x)$, where $y$ is the sequence of training data tokens and $x_{align}$ the sequence of source-side tokens that $y$ tokens are aligned to. Lower entropy indicates that the data is explained by a simpler word-to-word translation model that uses similar word translations irrespective of context.

Results are shown in Table 6. First, we observe that the complexity scores confirm the results reported by Zhou et al. (2020), with smaller-scale differences due to the fact that we mix both original data and pseudo-labeled data. BPE-DROPOUT performs best on smoothness w.r.t. synthetic noise: it outperforms all methods by a large margin on robustness, and by a smaller margin on consistency. This is not the case on data with natural noise (GMEG), where the increased consistency effect is smaller w.r.t. the BASELINE model. On other metrics, BPE-DROPOUT has no effect on quality (BLEURT) and a minor negative effect on stability across re-training. BPE-DROPOUT is not only the only method that lowers stability, but also the only method that increases the complexity of the data compared to the baseline.

BT shows a data simplification effect, mirrored by increased stability when re-training. However, BT has a detrimental effect on robustness and consistency metrics. These results indicate that while back-translation and forward translation are typically seen as very similar methods, they have different properties. PLT(SAMPLE) performs very similarly to PLT(TRAIN): when compared with PLT(TRAIN), it leads to slightly more complex data, and slightly worse quality and inertia scores. PLT(TRAIN) shows the lowest complexity scores and the highest stability.

While stability and complexity correlate, not all methods that simplify the data improve smoothness; conversely, smoothness to synthetic noise can be improved significantly with complementary methods such as BPE-DROPOUT. We corroborate Niu et al. (2020) and find that synthetic and natural noise are different in nature and not all methods are equally effective on both types of noise.

# 7 CONCLUSION

This paper investigates pseudo-label training, a technique common to a number of methods for boosting NMT performance. We show that in addition to well-studied gains in generic translation quality, pseudo-label training induces several desirable stability-related properties, which we group under the term *inertia*. Empirically, these improvements are not tied to the use of unlabeled data (as in self-training) or the use of stronger teacher models (as in knowledge distillation) but are a consequence of the use of pseudo-labeled data itself. When compared with other methods designed to improve robustness in NMT, we observed that the effect on stability over re-training occurs only for those methods that simplify the training data. Based on these findings, we recommend using PLT with unlabeled data (à la self-training) when developing NMT models where inertia is important due to its benefits to model inertia and its use in addressing potential language coverage bias (Wang et al., 2021). In future work, we plan to investigate the interplay between PLT and different formulations of NMT (auto- vs. non-autoregressive MT) as well as potential negative side effects such as bias amplification (Renduchintala et al., 2021). Finally, developing automatic metrics to detect negative flips in NMT is an important task that has yet to be examined extensively and can help guide PLT techniques.

ACKNOWLEDGEMENTS

We thank Yi Zhang and Elman Mansimov for discussions on negative flips and Miguel Ballesteros, Surafel Lakew, Cuong Hoang, and anonymous reviewers for their comments and suggestions.

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

## A    TRAINING PARAMETERS

All models used in our experiments utilized the following set of hyperparameters. Training and development data was tokenized using the Sacremoses tokenizer.[4] Words were segmented using BPE (Sennrich et al., 2016b) with 32K operations. Source and target subwords shared the same vocabulary. Training segments longer than 95 tokens were removed.

The source embeddings, target embeddings, and the output layer's weight matrix are tied (Press & Wolf, 2017). Training is done on 8 GPUs with Sockeye 3's large batch training. It has an effective batch size of 327,680 tokens, a learning rate of 0.00113 with 2000 warmup steps and a reduce

---

[4]`https://github.com/alvations/sacremoses`

rate of 0.9, a checkpoint interval of 125 steps, and learning rate reduction after 8 checkpoints without improvement. After an extended plateau of 60 checkpoints, the 8 checkpoints with the lowest validation perplexity are averaged to produce the final model parameters.

Parameters for standard training:

```
'learning_rate_scheduler_type': 'inv-sqrt-decay', 'keep_last_params':
10, 'update_interval': 16, 'transformer_model_size': (512, 512),
'transformer_postprocess': ('dr', 'dr'), 'learning_rate_warmup': 2000,
'transformer_dropout_act': (0.1, 0.1),
'transformer_feed_forward_num_hidden': (2048, 2048),
'max_num_checkpoint_not_improved': 60, 'weight_init_xavier_factor_type':
'avg', 'optimized_metric': 'perplexity', 'cache_strategy': 'best',
'num_layers': (20, 2), 'use_cpu': False,
'checkpoint_improvement_threshold': 0.001, 'device_ids': [-1],
'learning_rate_reduce_num_not_improved': 8, 'initial_learning_rate':
0.06325, 'seed': 1, 'cache_metric': 'perplexity',
'gradient_clipping_type': 'abs', 'cache_last_best_params': 8,
'weight_init_scale': 3.0, 'dtype': 'float32', 'decode_and_evaluate':
500, 'max_seconds': 1036800, 'amp': True, 'keep_initializations': True,
'transformer_dropout_prepost': (0.1, 0.1),
'transformer_attention_heads': (8, 8), 'weight_tying_type':
'src_trg_softmax', 'learning_rate_reduce_factor': 0.9, 'loss':
'cross-entropy', 'horovod': True, 'num_embed': (512, 512),
'embed_dropout': (0.0, 0.0), 'transformer_preprocess': ('n', 'n'),
'encoder': 'transformer', 'loglevel_secondary_workers': 'ERROR',
'label_smoothing': 0.1, 'batch_size': 2500, 'learning_rate_t_scale':
1.0, 'batch_type': 'max-word', 'optimizer': 'adam',
'transformer_dropout_attention': (0.1, 0.1), 'decoder':
'ssru_transformer', 'min_num_epochs': 1, 'checkpoint_interval': 500,
'transformer_positional_embedding_type': 'fixed', 'lock_dir': '/data',
'gradient_clipping_threshold': -1.0, 'weight_init': 'xavier',
'no_hybridization': False, 'batch_sentences_multiple_of': 8,
'transformer_activation_type': ('relu', 'relu')
```

## B  DATASET

We trained enen↔de models on Paracrawl v9 (Bañón et al., 2020), WikiMatrix (Schwenk et al., 2021), WikiTitles (Bojar et al., 2018), and news commentary datasets (Barrault et al., 2019). For en↔ru we additionally added the UN v1.0 dataset (Ziemski et al., 2016). For en↔ja we used JParaCrawl (Morishita et al., 2020) instead of ParaCrawl v9 and additionally added the Japanese-English subtitles dataset (Pryzant et al., 2018).

| LP | Datasets | # parallel |
|---|---|---|
| en↔de | Paracrawl v9, WikiMatrix, WikiTitles, news commentary | 286 M |
| en↔ja | JParacrawl v2, WikiMatrix, WikiTitles, news commentary, Japanese-English subtitles | 17.2 M |
| en↔ru | Paracrawl, WikiMatrix WikiTitles, news commenatry, UN v1.0 | 34 M |

Table 7:  We trained our models on a subset of datasets from the WMT21 news task. Specifically, we used Paracrawl v9 (Bañón et al., 2020), WikiMatrix (Schwenk et al., 2021), WikiTitles (Bojar et al., 2018), news commentary, UN v1.0 dataset (Ziemski et al., 2016), JParaCrawl (Morishita et al., 2020) and the Japanese-English subtitles datasets (Pryzant et al., 2018).

| LP | Years | # parallel |
|---|---|---|
| en↔de | 2017-2020 | 9k |
| en↔ja | 2020 | 2k |
| en↔ru | 2017-2020 | 9k |

Table 8:  We used the WMT news test datasets from previous years as our development set.

## C    TRAINING CURVES

Here, we compare pseudo-label training with back-translation (BT). We find that pseudo-label training regularizes the models by controlling for over fitting. BT also regularizes the model, but it does not simplify the distribution to the extent PLT does, implying that controlling over-fitting is not a main factor for stability. Comparisons with other methods (i.e. BPE-dropout and PLT(sample)) show similar trends.

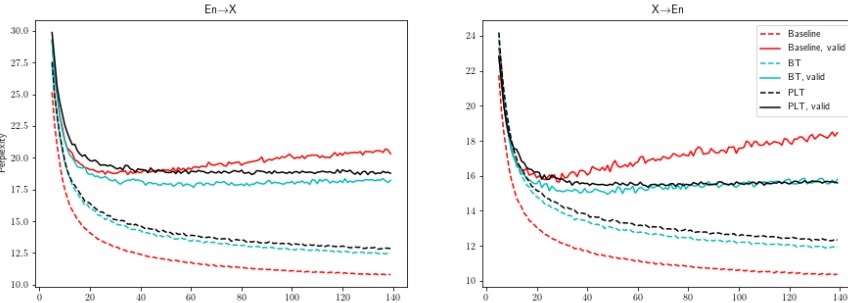

Figure 1: Comparisons of PLT(train) validation (solid lines) and training curves (dashed lines) against back-translation and baseline models. We find that in comparison, PLT is able to control over fitting on the training data. Back-translation also regularizes the model, but it does not simplify the distribution to the extent PLT does, implying that controlling over-fitting is not a main factor for stability.

## D    BERTSCORE

We provide quality scores using BERTScore (Zhang et al., 2020). In terms of generic quality, PLT provides improvements in quality consistent with earlier results using BLEU and BLEURT metrics (see Tables 2, 5, and 5). We also computed robustness metrics BERTScore:

$$\texttt{Robustness} \quad = \quad \text{BERTScore}(Y', Y_{ref}) - \text{BERTScore}(Y, Y_{ref}) \tag{3}$$

where $Y_{ref}$ stands for reference translations and $Y, Y'$ are translations of a *clean/noisy* versions of the test set (e.g., one with introduced misspellings).

| LP | Setting | Teacher | Student (Original + PL) | BLEU | BERTScore | Misspellings | | GMEG |
|---|---|---|---|---|---|---|---|---|
| | | | | | | Rob | Const | Const |
| en→de | ALLDATA | – | 286M + 0 | $27.83_{\pm1.1}$ | $0.860_{\pm0.002}$ | $\mathbf{-0.016}_{\pm0.001}$ | $73.9_{\pm0.9}$ | 83.1 |
| | BASELINE | – | 143M + 0 | $27.62_{\pm1.1}$ | $0.860_{\pm0.002}$ | $-0.016_{\pm0.001}$ | $73.1_{\pm0.9}$ | 82.8 |
| | PLT(TRAIN) | 143M | 143M + 143M (Train) | $27.86_{\pm1.1}$ | $\mathbf{0.861}_{\pm0.002}$ | $-0.017_{\pm0.001}$ | $76.5_{\pm0.8}$ | 85.1 |
| | PLT(UL) | 143M | 143M + 143M (UL) | $\mathbf{28.07}_{\pm1.1}$ | $0.861_{\pm0.002}$ | $-0.018_{\pm0.001}$ | $\mathbf{76.8}_{\pm0.8}$ | **85.3** |
| en→ru | ALLDATA | – | 34M + 0 | $25.71_{\pm1.1}$ | $0.856_{\pm0.002}$ | $-0.018_{\pm0.001}$ | $66.4_{\pm1.1}$ | 79.3 |
| | BASELINE | – | 17M + 0 | $25.08_{\pm1.1}$ | $0.854_{\pm0.002}$ | $-0.019_{\pm0.001}$ | $64.9_{\pm1.0}$ | 78.4 |
| | PLT(TRAIN) | 17M | 17M + 17M (Train) | $\mathbf{26.16}_{\pm1.1}$ | $0.857_{\pm0.002}$ | $-0.017_{\pm0.001}$ | $\mathbf{70.2}_{\pm1.0}$ | 81.5 |
| | PLT(UL) | 17M | 17M + 17M (UL) | $25.87_{\pm1.1}$ | $\mathbf{0.857}_{\pm0.002}$ | $-0.017_{\pm0.001}$ | $70.1_{\pm1.0}$ | **82.2** |
| en→ja | ALLDATA | – | 17.2M + 0 | $23.62_{\pm0.8}$ | $0.836_{\pm0.001}$ | $-0.020_{\pm0.001}$ | $59.3_{\pm1.1}$ | 72.6 |
| | BASELINE | – | 8.6M + 0 | $22.82_{\pm0.9}$ | $0.833_{\pm0.001}$ | $-0.020_{\pm0.001}$ | $59.3_{\pm1.1}$ | 71.5 |
| | PLT(TRAIN) | 8.6M | 8.6M + 8.6M (Train) | $24.63_{\pm0.9}$ | $0.839_{\pm0.001}$ | $\mathbf{-0.020}_{\pm0.001}$ | $\mathbf{64.5}_{\pm1.1}$ | **75.9** |
| | PLT(UL) | 8.6M | 8.6M + 8.6M (UL) | $\mathbf{24.59}_{\pm0.9}$ | $\mathbf{0.839}_{\pm0.001}$ | $-0.021_{\pm0.001}$ | $\mathbf{64.5}_{\pm1.1}$ | **75.9** |
| de→en | ALLDATA | – | 286M + 0 | $32.32_{\pm1.1}$ | $0.953_{\pm0.001}$ | $\mathbf{-0.012}_{\pm0.001}$ | $77.4_{\pm1.0}$ | - |
| | BASELINE | – | 143M + 0 | $32.36_{\pm1.1}$ | $0.953_{\pm0.001}$ | $-0.012_{\pm0.001}$ | $76.9_{\pm1.0}$ | - |
| | PLT(TRAIN) | 143M | 143M + 143M (Train) | $\mathbf{32.93}_{\pm1.1}$ | $\mathbf{0.953}_{\pm0.001}$ | $-0.013_{\pm0.001}$ | $80.0_{\pm0.8}$ | - |
| | PLT(UL) | 143M | 143M + 143M (UL) | $32.50_{\pm1.1}$ | $0.953_{\pm0.001}$ | $-0.013_{\pm0.001}$ | $\mathbf{80.2}_{\pm0.9}$ | - |
| ru→en | ALLDATA | – | 34M + 0 | $36.39_{\pm1.2}$ | $0.956_{\pm0.001}$ | $-0.004_{\pm0.000}$ | $85.6_{\pm0.8}$ | - |
| | BASELINE | – | 17M + 0 | $36.26_{\pm1.2}$ | $0.956_{\pm0.001}$ | $-0.005_{\pm0.000}$ | $84.2_{\pm0.9}$ | - |
| | PLT(TRAIN) | 17M | 17M + 17M (Train) | $\mathbf{37.00}_{\pm1.2}$ | $0.957_{\pm0.001}$ | $-0.004_{\pm0.000}$ | $\mathbf{87.6}_{\pm0.7}$ | - |
| | PLT(UL) | 17M | 17M + 17M (UL) | $36.85_{\pm1.12}$ | $\mathbf{0.957}_{\pm0.001}$ | $\mathbf{-0.004}_{\pm0.000}$ | $87.0_{\pm0.7}$ | - |

Table 9: Training data sizes and performance scores for PLT/Baseline models. Quality is measured with BLEU and BERTScore on the WMT21 news test set. Smoothness is measured as robustness and consistency to synthetic (Misspellings) and natural (GMEG) noise. GMEG scores are computed as the average over four reference corrections. Robustness measures changes in translation quality w.r.t input variations, while Consistency measures translation *changes* alone.

| LP | Teacher | | | PLT | | | | | |
|---|---|---|---|---|---|---|---|---|---|
| | Arch. | BLEU | BERTScore | BLEU | BERTScore | Stability | Const(GMEG) | Rob(Missp) | Const(Missp) |
| en→X | – | – | – | 25.12 | 0.845 | 62.20 | 77.57 | -0.018 | 65.77 |
| | 20:1 | 24.08 | 0.845 | 25.54 | 0.850 | 72.67 | 79.93 | -0.018 | 69.72 |
| | 20:2 | 25.12 | 0.849 | 26.12 | 0.852 | **74.10** | 80.72 | **-0.018** | **70.38** |
| | 20:4 | **25.63** | **0.849** | **26.27** | 0.852 | 73.68 | **80.87** | -0.018 | 70.05 |
| X→en | – | – | – | 28.12 | 0.937 | 61.91 | – | -0.009 | 80.53 |
| | 20:1 | 26.74 | 0.932 | 28.30 | 0.936 | 73.15 | – | -0.008 | 83.56 |
| | 20:2 | 28.12 | **0.937** | 29.56 | **0.939** | **73.96** | – | -0.009 | 83.80 |
| | 20:4 | **28.40** | 0.937 | **29.94** | 0.937 | 73.86 | – | **-0.008** | **83.99** |

Table 10: PLT models using teachers of varying quality as measured by BERTScore (averages over the three LPs in each direction). We find that teacher quality correlates with quality of PLT; however, weaker teachers can still improve student quality. In terms of inertia properties, these are preserved regardless of teacher quality. Note X→en averages for robustness and consistency to misspelling include only de,ru→en.

| LP | Setting | $C(d)\downarrow$ | BLEU | BERTScore | Stability | Const(GMEG) | Rob(Missp) | Const(Missp) |
|---|---|---|---|---|---|---|---|---|
| en→X | BASELINE | 3.74 | 25.12 | 0.849 | 62.20 | 77.57 | -0.018 | 65.77 |
| | BT | 3.64 | 25.96 | 0.851 | 65.36 | 77.16 | -0.021 | 64.83 |
| | BPE-DROPOUT | 3.90 | 24.82 | 0.847 | 61.70 | 78.69 | **-0.011** | **71.86** |
| | PLT(SAMPLE) | 3.56 | 25.96 | 0.852 | 71.97 | 80.41 | -0.018 | 69.87 |
| | PLT(TRAIN) | **3.54** | **26.12** | **0.852** | **74.10** | **80.73** | -0.018 | 70.38 |
| X→en | BASELINE | 3.12 | 28.12 | 0.936 | 61.91 | – | -0.009 | 80.53 |
| | BT | 3.01 | 28.88 | 0.938 | 64.91 | – | -0.010 | 80.00 |
| | BPE-DROPOUT | 3.41 | 28.19 | 0.937 | 61.73 | – | **-0.006** | **84.96** |
| | PLT(SAMPLE) | 2.98 | 29.41 | 0.939 | 72.69 | – | -0.009 | 83.70 |
| | PLT(TRAIN) | **2.94** | **29.56** | **0.939** | **73.96** | – | -0.009 | 83.80 |

Table 11: Performance and model inertia with PLT versus other methods (averages over the three LPs in each direction). Stability to model updates is computed w.r.t. to random seed variation in student models. X→en averages for robustness and consistency to misspellings involve de,ru→en.

