# OpenReview forum: "Pseudo-label Training and Model Inertia in Neural Machine Translation"
_ICLR.cc/2023/Conference — ICLR 2023 poster_

### Official Review · Reviewer_Ypio · 2022-10-23

**Confidence:** 4
**Correctness:** 3
**Technical Novelty And Significance:** 3
**Empirical Novelty And Significance:** 2
**Recommendation:** 5

**Clarity, Quality, Novelty And Reproducibility:**


* The quality is not on par, as I am expecting better explanation on the implication and relevant suggestions to improve NMT models from these analysis
* Novelty is fine.
* Reproducibility: there is no code available.

**Strength And Weaknesses:**

Strength

* The paper seems quite clear and easy to understand.
* The experiment results are decent

Weakness

* Lack various comparisons with related methods that try to do the same, such as robustness training against adversarial attack.
* The paper doesn't conclude or provide any suggestions or implication from the results so that it is hard to find the key takeaways from the paper, and it's difficult to imagine what we can do about the knowledge from the paper. For example, the paper should show experiments that key learning from analysis of model inertia helps us develop better model that perform better. I do not see any of such indication in the paper. And thus, I find it difficult to find this useful.
* Missing citations of some pseudo-label training forward translation methods, such as data diversification (Nguyen et al., 2020, NeuIPS)


**Summary Of The Paper:**

The paper essentially reframes NMT models’ stability to model updates and input perturbations as *model inertia* and conduct various analysis experiments on it in the  pseudo-label training setups, aka forward-translation distillation. The paper shows that quality gains in different methods are due to the use of pseudo-labeled data in training.

**Summary Of The Review:**

Overall, I think the paper is written with clear presentation. However, I think the analysis of model inertia is largely expected, and does not provide key learnings as there is no suggestions or conclusions on how to improve NMT model with this knowledge.

---

> ### Author Response · Authors · 2022-11-18
> **Response to reviewer Ypio**
>
> Thank you for taking the time to read our manuscript and providing thoughtful comments and suggestions! We’ve addressed your comments below.
>
> 1. In our experiments, we compared against BPE-dropout as a method to improve NMT model robustness to small input variations that may come from user inputs. BPE-dropout is a method that’s known to improve robustness but not decrease complexity and by comparing PLT methods against this method, we can compare what role the distribution simplification plays. Comparing against adversarial training and seeing if similar distribution simplification effects are present is an interesting idea and one direction for future work.
> 2. A key learning from our research is that pseudo-label training improves NMT model stability to input perturbations and across incremental model updates. While pseudo-label training has been studied in the past, earlier works have focused on the improvements to quality. In particular, in autoregressive NMT, pseudo-label training is not used as widely as say back-translation. Our aim is to highlight that researchers and practitioners should use pseudo-label training when training NMT models as a way to increase model stability. We’ve highlighted this in our introduction and conclusion to make this more explicit. We’ve also highlighted some areas for future work such as work on metrics for quantifying negative flips in NMT which we noted is trickier and less well studied than in classification tasks.
> 3. Thank you for pointing out this missing citation. Data diversification is an important aspect of pseudo-label training as noted by Nguyen et al., 2020. We’ve added this citation.

---

### Official Review · Reviewer_ZSXu · 2022-10-24

**Confidence:** 4
**Correctness:** 3
**Technical Novelty And Significance:** 2
**Empirical Novelty And Significance:** 2
**Recommendation:** 5

**Clarity, Quality, Novelty And Reproducibility:**

The experiments are carefully designed, e.g. they employed the test set with the source-original ones, which have the results be more reliable.
When discussing the model stability or robustness, the authors mainly report automation evaluation metrics and I was wondering how the training curve or convergency has changed in the pseudo-label training. Do you have the loss curve plots for each model and see any trends?
The paper looks a good analytic report but does not come up with a novel approach or proposal. It is okay in this format, though I would like to learn how to further improve robustness in NMT models based on these findings.

**Strength And Weaknesses:**

Strengths
- Extensive experimental results to assess the NMT models and pseudo-label training in different scores
- These analytic report would be useful to understand the impact of pseudo-label training

Weaknesses
- Lots of useful observation from the extensive experiments, however, there is few proposed approaches. Or, theoretical proof would be appreciated.

**Summary Of The Paper:**

The authors do research on pseudo-label training that is also known as knowledge distillation or teacher-student training, which transfers task knowledge learned from a larger model to a smaller model by letting the smaller model mimic the teacher's outputs rather the gold labels during the training. They study "model inertia effects" of the pseudo-label training and discuss how to address the robustness training in NMT models. The NMT models are trained on six different language directions and assessed in a variety of automation metrics to capture robustness and consistency, in addition to the n-gram match accuracy like BLEU scores.

**Summary Of The Review:**

The paper provides substantial experimental results and discuss inertia effect in the different MT task settings. The paper came up with a lots of observation and suggestions in a carefully designed experiments, however, it does not end up with any novel approaches or concrete conclusions for further improvement. We probably would like to have more theoretical proof or explanations behind the observations.

---

> ### Author Response · Authors · 2022-11-18
> **Response to reviewer ZSXu**
>
> Thank you for taking the time to read our paper and for your comments and suggestions.
>
> In our work, we explored several variations of pseudo-label training (forward translation and knowledge distillation) and found that in addition to well documented improvements to model quality, there is also a benefit to model stability to input perturbations and across incremental model updates which has not been focused on in earlier works. In particular, in autoregressive NMT, pseudo-label training is less widely used than other techniques and our recommendation for practitioners and researchers going forward is to incorporate pseudo-label training whenever model inertia is an important factor as it doesn’t sacrifice quality. We’ve highlighted this in the introduction and conclusions.
>
> We appreciate your suggestion for including the training curves and have added them to the paper in Appendix C. We’ve compared pseudo-label training with backtranslation which we've shown to not simplify the training distribution to the extent pseudo-label training does. We found that pseudo-label training regularizes the models by reducing overfitting. BT also regularizes the model, but it does not simplify the distribution implying that controlling overfitting is not a major factor for stability.

---

### Official Review · Reviewer_fgwu · 2022-10-24

**Confidence:** 3
**Correctness:** 4
**Technical Novelty And Significance:** 3
**Empirical Novelty And Significance:** 2
**Recommendation:** 8

**Clarity, Quality, Novelty And Reproducibility:**

The paper is very clearly written, formulas look clean and consistent, although using an indicator function in Eq. 1 seems odd (but not wrong) to me. Experimental settings are described sufficiently, and the reported numbers are effective to support the claims. Baseline BLEU scores in Table 2 seem to be in the right ballpark.

**Strength And Weaknesses:**

The paper makes a pretty straightforward point: Pseudo-labels improve the robustness of NMT inference and the stability of NMT training. This point may not be overly surprising to MT practioners. But that doesn't mean that it is a useful contribution to demonstrate this as thoroughly and clearly as this paper.

I don't see any major flaws, but here are some minor points:
- Where applicable, significance tests in Sec. 5.1 would be good.
- I'd encourage the authors to add some citations on faithfulness / robustness in NMT. Things like https://arxiv.org/pdf/2005.12398.pdf also come to mind, as well as the renewed popularity of the lottery ticket hypothesis.
- "NMT lacks a reliable automatic segment-level quality metric" There is evidence that neural MT metrics (BLEURT, COMET, ...) are more reliable on the sentence level than BLEU.

**Summary Of The Paper:**

Pseudo-labels are widely used in practical NMT, either in a classical (sequence-level) KD scenario or for forward translation. The paper demonstrates that both forms of pseudo-labels improve the sanity of NMT by showing that models are both more robust against input perturbations and less prone to variations between training runs due to random initialization. Experiments on 6 language pairs using existing robustness measures and new model stability metrics as well as human evals support their claims convincingly.

**Summary Of The Review:**

A solid and sober paper, but not the most exciting one.

---

> ### Author Response · Authors · 2022-11-18
> **Response to reviewer fgwu**
>
> Thank you for your review and suggestions! We’ve addressed your comments and concerns below.
>
> 1. We’ve added significance numbers to the tables in section 5.1.
>
> 2. While COMET and BLEURT are better correlated with human evaluation of quality, as has been shown by earlier works ([Kocmi et al, 2021](https://aclanthology.org/2021.wmt-1.57/), [Mathur et al, 2021](https://aclanthology.org/2020.wmt-1.77/)) , segment-level correlation is usually much lower than document/system-level correlation with human perceptions of quality. Hence, at a segment level, we still do not have reliable automatic metrics. We’ve added this clarification and citations to our manuscript. That being said, since neural metrics are more reliable on a system level, we’ve included BLEURT scores in our system evaluations of quality and updated our measurement of robustness to use BLUERT instead of BLEU scores. The main conclusions of our paper were unchanged.
>
> 3. Lastly, thank you for pointing out the missing citations. We’ve included the citations on faithfulness and robustness.

---

### Official Review · Reviewer_ixEF · 2022-10-26

**Confidence:** 4
**Correctness:** 3
**Technical Novelty And Significance:** 2
**Empirical Novelty And Significance:** 2
**Recommendation:** 5

**Clarity, Quality, Novelty And Reproducibility:**

**Clarity**
The paper is mostly clear. It can be improved if it can give a more detailed formulation of each setting in Section 5 since this part has many repetitive and similar settings (e.g., PLT (XX), PLT-delta(XX)).

**Quality**
This is an OK paper and indeed provides some insights for the line of research. However, it can be a better paper if the authors can further investigate the core factors that lead to model inertia.

**Novelty**
Both PLT and the metrics to analyze model inertia are not novel. But combining these has some novelty.

**Reproducibility**
The experimental settings are clear for reproducing.

**Strength And Weaknesses:**

Strength:
Overall, PLT is a simple and effective method for improving the model performance of NMT. The understanding of PLT is an important topic. This paper gives some insights that might be useful.

Weaknesses:
1. The main weakness is that the analysis of the model inertia just focuses on the evaluation of model results but ignores the analysis of the pseudo-labels themselves. Which kind of characteristics of the pseudo-labels affect model inertia? For example, previous works have shown that NMT models are more likely to make frequent predictions, monotonic translations, and so on. In other words, why simplifying the training data can bring such benefits? The core questions are not well answered.
2. After carefully reading the paper, I still cannot find how can the paper findings guild future research. Could we design better PLT methods based on the findings, and how? Or could we better train NMT models with PLT?
3. I strongly suggest the authors include a comparison of non-autoregressive models, which might attract more attention from the community.
4. In 4.1, COMET or BERTScore could be a better choice for evaluating the robustness or consistency.

---
Thank you for the response that has alleviated my part concerns. I would like to increase my score a bit. However, similar to Reviewers ypio and zsxu, how can the paper's findings guild future research is still unclear which limits me from giving a higher rating.

**Summary Of The Paper:**

This paper broadens the understanding of pseudo-label training (PLT) in neural machine translation (NMT). It finds that PLT can not only improve model performance but also enhance model stability.

**Summary Of The Review:**

This is an interesting paper that investigates the relationship between PLT and model inertia. However, it does not provide enough insights to guide future research. Therefore, I recommend rejection at this moment.

---

> ### Author Response · Authors · 2022-11-18
> **Response to reviewer ixEF**
>
> Thank you for taking the time to read our paper and providing insightful comments! We’ve addressed your specific concerns below:
>
> 1. We’ve looked at the effect that the pseudo-labels have on the training distribution and training curves (see Appendix C or response to Reviewer ZSXu). We’ve compared against several different methods, like BPE-dropout, which is known to improve robustness but not simplify the training distribution, and back-translation, which is known to improve quality and likewise not simplify the distribution. We find that pseudo-label training regularizes the model (as evident from the training curves) *and* simplifies the distribution (as evident from Table 6 in the paper). We find that the distribution simplification effect is strongly correlated with increased stability and consistency.
>
> 2. The aim of our work was to investigate pseudo-label training in autoregressive NMT. While pseudo-label training is more popular in non-autoregressive NMT, it is less widely used in autoregressive machine translation. Prior work highlights the benefits to quality. We extend earlier works by highlighting another benefit: pseudo-label training also improves model stability to input perturbations and incremental updates. These are important features for downstream users of the model outputs. Based on our observations, we would recommend that in training autoregressive NMT models, one should incorporate pseudo-label training when model inertia is an important consideration,as it does not sacrifice quality. (We’ll make this recommendation more explicit in the paper.) We leave for future work the development of metrics to quantify negative flips in NMT (which we note is much trickier and less studied than in classification tasks) which ties in to the issue of students generating more frequent predictions, monotonic translations etc.
>
> 3. In our experiments, we sought to compare against state-of-the-art methods of machine translation which are the autoregressive NMT models and demonstrated that pseudo-label training has benefits even for these models. You’re right that since pseudo-label training is used in non-autoregressive MT, we expect that our research would be of interest to that researchers in that field as well.
>
> 4. We’ve updated our manuscript to include BLEURT scores for our system-level evaluations and used the BLEURT score for robustness since that compares changes in quality relative a reference translations. In terms of consistency, we’ve left it as using BLEU scores since we are more interested in the lexical similarity of the outputs rather than their semantic similarity.

---

> > ### Comment · Reviewer_ixEF · 2022-11-19
> > **Could you inculde BERTScore?**
> >
> > Thank you for the response. Could you show the BERTScores of the response 4?

---

> > > ### Author Response · Authors · 2022-11-21
> > > **BERTScores**
> > >
> > > We are happy to provide BERTScores for our experiments.
> > > Below, we've provided average quality scores over en$\rightarrow$X
> > > and X$\rightarrow$en language pairs. We find that our main conclusions are unchanged
> > > and plan on updating the camera ready version with confidence intervals
> > > and robustness metrics.
> > >
> > > In terms of generic quality, PLT provides improvements in quality inline with earlier
> > > results using BLEU and BLEURT metrics (see Table 1).
> > >
> > > Table 1: Generic quality on newstest2021 of PLT versus baseline models using BERTScore
> > >
> > > |LP    | Setting     |BERTScore|
> > > |:-----|:------------|--------:|
> > > |      | AllData    |0.850    |
> > > |      | Baseline          |0.849    |
> > > |en$\rightarrow$X | PLT(Train) |0.852    |
> > > |      | PLT(UL)   |**0.852**|
> > > |------|----------|--------------|
> > > |      | AllData    |0.937    |
> > > |      | Baseline   |0.936    |
> > > |X$\rightarrow$en | PLT(Train) |0.939   |
> > > |      | PLT(UL)   |**0.939**|
> > >
> > > In terms of teacher quality, we find that teacher quality still correlates with
> > > PLT student quality and that weaker teachers also provide a benefit (see Table 2).
> > >  This is corroborates our earlier results using BLEU and BLEURT metrics.
> > >
> > > Table 2: Experiments varying teacher quality
> > >
> > > |LP    | Teacher | Teacher BERTScore| Student BERTScore |
> > > |:-----|:--------|:------------:|-----------------:|
> > > |                |20:1|0.845    | 0.850|
> > > |en$\rightarrow$X|20:2|0.849    | 0.852|
> > > |                |20:4|**0.849**   | **0.852**|
> > > |------|---------|--------|---------|
> > > |                |20:1|0.936    |0.936|
> > > |X$\rightarrow$en|20:2|**0.937**    |**0.939**|
> > > |                |20:4|0.937    |0.937|
> > >
> > > Finally, in comparing with other methods that do not simplify the training
> > > distribution, we find that the BERTScore mirrors our earlier results using
> > > BLEU and BLEURT metrics (see Table 3).
> > >
> > > Table 3: Experiments comparing PLT(Train) with methods that do not simplify the distribution.
> > >
> > > |LP    | Setting     |BERTScore|
> > > |:-----|:------------|--------:|
> > > |      | Baseline    |0.849    |
> > > |      | BT          |0.851    |
> > > |en$\rightarrow$X | BPE-Dropout |0.847    |
> > > |      | PLT(Sample)  |0.852    |
> > > |      | PLT(Train)   |**0.852**|
> > > |------|-------------|---------|
> > > |      | Baseline    |0.936    |
> > > |      | BT          |0.938    |
> > > |X$\rightarrow$en | BPE-Dropout |9.937    |
> > > |      | PLT(Sample)  |0.939    |
> > > |      | PLT(Train)   |**0.939**|

---

### Decision · Program_Chairs · 2023-01-20

**Decision:**

Accept: poster

**Justification For Why Not Higher Score:**

- the paper is clear and shows a good experimental investigation with a strong analysis of the results.

**Justification For Why Not Lower Score:**

- the paper could communicate recommendations for future research and would benefit from analyzing if PL-training induces a change in the style of translations.

**Metareview: Summary, Strengths And Weaknesses:**

This paper studies the impact of training on pseudo labels (PL) in neural machine translation (NMT). In addition to improving model accuracy, pseudo labels are shown to also enhance training stability and robustness to input perturbations.

Strengths
- extensive experiments, clear analysis of PL training in NMT.

Weaknesses:
- the paper does not delineate the implications of its conclusions and how future research could benefit from it.
- the paper relies on BLEU and BLEURT for evaluation but does not evaluate if PL trained models produce different translation, e.g. simpler, more mechanical translation, see ixEF review.


**Note From Pc:**

if the above contains the word "oral" or "spotlight" please see: "oral" presentation means -> notable-top-5% and "spotlight" means -> notable-top-25%. As stated in our emails, we are disassociating presentation type from AC recommendations